# Changes in Multimorbidity and Polypharmacy Patterns in Young and Adult Population over a 4-Year Period: A 2011–2015 Comparison Using Real-World Data

**DOI:** 10.3390/ijerph18094422

**Published:** 2021-04-21

**Authors:** Sara Mucherino, Antonio Gimeno-Miguel, Jonas Carmona-Pirez, Francisca Gonzalez-Rubio, Ignatios Ioakeim-Skoufa, Aida Moreno-Juste, Valentina Orlando, Mercedes Aza-Pascual-Salcedo, Beatriz Poblador-Plou, Enrica Menditto, Alexandra Prados-Torres

**Affiliations:** 1CIRFF, Center of Drug Utilization and Pharmacoeconomics, University of Naples Federico II, 80131 Naples, Italy; sara.mucherino@unina.it (S.M.); valentina.orlando@unina.it (V.O.); 2Department of Pharmacy, University of Naples Federico II, 80131 Naples, Italy; 3EpiChron Research Group, Aragon Health Sciences Institute (IACS), IIS Aragón, Miguel Servet University Hospital, 50009 Zaragoza, Spain; agimenomi.iacs@aragon.es (A.G.-M.); jcarmona@iisaragon.es (J.C.-P.); franciscagonzalezrubio@gmail.com (F.G.-R.); ignacio.ioakim@hotmail.es (I.I.-S.); aidamorenoj@gmail.com (A.M.-J.); maza@salud.aragon.es (M.A.-P.-S.); bpoblador.iacs@aragon.es (B.P.-P.); sprados.iacs@aragon.es (A.P.-T.); 4Health Services Research on Chronic Patients Network (REDISSEC), ISCIII, 28029 Madrid, Spain; 5Drug Utilization Work Group, Spanish Society of Family and Community Medicine (SemFYC), 28004 Madrid, Spain; 6Vaksinasjonssenter BSN, Bydel Søndre Nordstrand, Oslo Kommune, 1252 Oslo, Norway

**Keywords:** multimorbidity, polypharmacy, chronic diseases, real-world data, epidemiology

## Abstract

The pressing problem of multimorbidity and polypharmacy is aggravated by the lack of specific care models for this population. We aimed to investigate the evolution of multimorbidity and polypharmacy patterns in a given population over a 4-year period (2011–2015). A cross-sectional, observational study among the EpiChron Cohort, including anonymized demographic, clinical and drug dispensation information of all users of the public health system ≥65 years in Aragon (Spain), was performed. An exploratory factor analysis, stratified by age and sex, using an open cohort was carried out based on the tetra-choric correlations among chronic diseases and dispensed drugs during 2011 and compared with 2015. Seven baseline patterns were identified during 2011 named as: mental health, respiratory, allergic, mechanical pain, cardiometabolic, osteometabolic, and allergic/derma. Of the epidemiological patterns identified in 2015, six were already present in 2011 but a new allergic/derma one appeared. Patterns identified in 2011 were more complex in terms of both disease and drugs. Results confirmed the existing association between age and clinical complexity. The systematic associations between diseases and drugs remain similar regarding their clinical nature over time, helping in early identification of potential interactions in multimorbid patients with a high risk of negative health outcomes due to polypharmacy.

## 1. Introduction

Polypharmacy is referred to as the concurrent use of multiple drugs, and it can be the natural consequence of multimorbidity, more often intended as the coexistence of two or more chronic diseases [1]. However, inappropriate polypharmacy increases the risk of unnecessary drug use, potential drug–drug and drug–disease interactions, and adverse drug reactions (ADRs) [2,3], representing an economic and public health issue related to the quality and efficiency of health care [4,5]. The lack of development of specific care models for this population aggravates multimorbidity and polypharmacy [6].

Large-scale population studies based on real-world data represent an excellent opportunity to analyze the complexity of drug prescribing and clinical conditions and allow us to investigate the existence of systematic associations among drugs and diseases [7,8,9,10]. Factor analysis can improve the understanding of multimorbidity and polypharmacy in a real-world context. In 2015, we conducted a study that revealed the existence of systematic associations among chronic diseases and dispensed drugs, identifying up to six patterns of multimorbidity and polypharmacy [11]. Hence, this study aims to compare the baseline epidemiological patterns of multimorbidity and polypharmacy of the EpiChron Cohort in 2011 with those published in 2015 and to describe the clinical evolution of the clinical clusters identified.

## 2. Materials and Methods

### 2.1. Design, Study Population, and Variables

We performed an observational, cross-sectional study in the EpiChron Cohort [12]. This cohort includes the anonymized demographic, drug dispensation and clinical information of 98% of users of the public health system in Aragon, Spain (about 1.3 million inhabitants). We collected data from 2011 and compared them with previously published data from 2015 [11] in order to make a 4-year comparison.

The study population included all the subjects living in the Aragon region up to 65 years of age who were users of the public health system. Patients aged 65 and older were excluded from the study to allow for focus on young and adult populations for reasons already explained [11]. We stratified the population by sex and into three age groups: 0–14, 15–44 and 45–65 years, as for the previous analysis to compare the same age groups. For each subject, we analyzed all the diagnoses of chronic diseases from primary care and hospital electronic health records and all dispensed drugs from pharmacy billing records during 2011.

Diagnoses were coded initially based, first on the International Classification of Primary Care (ICPC) and then converted to codes of the International Classification of Diseases 9th Revision (ICD-9). Finally, they were grouped in the Expanded Diagnostic Clusters (EDC) of the ACG System (version 11.0, The Johns Hopkins University, Baltimore, MD, USA). We included in the analysis all 114 diseases classified as chronic by Salisbury et al. [13] and coded in binary format (i.e., presence/absence of the disease). As in the 2015 study, we also included rhinitis, following the World Health Organization (WHO) indications [14], and acute lower respiratory tract infection, as it can generate chronic sequelae. We classified dispensed drugs according to their Anatomical Therapeutic Chemical (ATC) code at the third level and included chronic and acute drug dispensation with a prevalence of at least 3% in 2015. The Clinical Research Ethics Committee of Aragón (CEICA) approved the study (ethical approval code: PI18/041) and waived the requirement for patient consent, since data of the EpiChron Cohort are anonymized, and no interventions on individuals were performed.

### 2.2. Statistical Analyses

As we used an open cohort, we performed a descriptive analysis of both 2011 and 2015 populations by describing demographic and clinical information expressed as frequencies, means, standard deviations (SD), and medians. We compared differences between patient characteristics using the chi-squared test for categorical variables or the unpaired *t*-test for numerical variables, as appropriate, considering statistically significant a *p* value < 0.05. Patients’ characteristics compared were age, area of living, immigrant status, deprivation index and number of chronic diseases, multimorbidity, and number of drugs related to the reference year. The deprivation index is strictly related to the census section of subjects, which represents the degree of deprivation from the lowest (Q1) to the highest (Q4) of the administrative health area to which it belongs.

An exploratory factor analysis was performed to identify multimorbidity and polypharmacy patterns according to a correlation matrix to decide which diagnoses and dispensed drugs comprised each pattern. Tetra-choric correlation matrices were used due to the dichotomous nature of both chronic diagnoses and administrated medicines. We performed factor extraction based on the principal factor method. We also applied an oblique rotation (Oblimin) to facilitate factor interpretation.

Scree plots were used to decide the number of factors to extract in each group. To determine which codes formed each pattern, we included those with scores >0.30 for each factor. This is the threshold factor loading traditionally used when deciding whether to accept a variable as belonging to a factor [11]. Nonetheless, as done in the previous study, EDCs and ATC codes with scores between 0.25 and 0.30 were included in a factor if considered relevant in the clinical explanation of the pattern.

As done in the previous work [11], we included EDCs with a prevalence >1–2% and ATC codes with a prevalence >3–5% in each age and sex group. Some ATCs with lower prevalence were also covered based on their potential relevance for interactions or side effects. The inclusion and exclusion criteria of EDCs and ATC codes used for each sex and age group were the same, explicitly explained in the 2015 study [11]. We used this prevalence threshold to increase the epidemiological interest of the study, and for statistical reasons regarding collinearity amongst some of the studied variables. The order of factors depends on the prevalence of its components. ATCs and EDCs with higher prevalence values will be identified in the first factors.

We evaluated sample adequacy using the Kaiser–Meyer–Olkin (KMO) test. We only considered values >0.60 as acceptable. Moreover, we calculated the proportion of cumulative variance as a measure of the model’s goodness-of-fit. This measurement describes the data variability explained by the patterns. We conducted all statistical analyses in STATA (version 12.0, StataCorp LLC, College Station, TX, USA).

### 2.3. Differences in the Clinical Patterns Evaluation Process

Once we obtained the data, the clinical nature of the patterns identified, and the comparability of the patterns over the 4-year period analyzed, we identified the presence of potential interactions between diseases and drugs within the patterns and the substantial differences observed. The associations found in each pattern were independently reviewed by three pharmacists (E.M., V.O., and S.M.) and seven physicians (F.G.R., M.A.S., A.M.J., A.J.M., I.I.S., J.C.P. and A.P.T.) from the research team. Subsequently, a consensus meeting was held to discuss and analyze the differences that existed at the turn of four years. We retained the names of the clusters given in the previous published study with 2015 data, wherever possible, to ensure a better reading of the difference over the years. Finally, the differences observed between 2011 and 2015 were compared with existing literature.

## 3. Results

Subjects identified up to 65 years old in the Aragon region were 1,000,390 during 2011 and 887,572 during 2015. Comparison and description of demographic and clinical characteristics of the two study populations are shown in Table 1 for women and Table 2 for men. Firstly, for both the years 2011 and 2015, we detected a statistically significant increase in the number of drugs and chronic conditions for both sexes as age increases.

### 3.1. Comparison of Multimorbidity and Polypharmacy Patterns

All the six epidemiological patterns identified in 2015 were also maintained during 2011, named as respiratory, mental health, cardiometabolic, endocrinological, osteometabolic, and mechanical pain. In addition, a new one appeared in 2011 mainly in younger age groups, recognized as an allergic/derma factor. Comparison of multimorbidity and polypharmacy patterns are detailed in Table 3.

#### 3.1.1. Girls Aged 0–14 Years

Scree plot identified four factors during 2011 versus three during 2015 (Table 4). Factors identified in 2015 in girls in this age group were generally already present in 2011, but with the addition of an allergic/derma component recognized in 2011 and not maintained in 2015. In contrast, the first factor of 2015 was identified as respiratory/acute infection due to the presence of acute lower respiratory tract infection conditions and anti-infectives, corticosteroids, antifungals, and antibiotics. Second factors were similar in both years, having respiratory/asthmatic character due to the equal presence of asthma but differed for drugs-related such as adrenergics and corticosteroids for 2011 and antihistamines and decongestants for 2015. The third factor, the allergic one, with allergic rhinitis and antihistamines and decongestants, appeared only in 2011 in the pediatric population. The last factor identified as mental health remained unchanged over the years due to the presence of developmental disorders and psychosocial disorders of childhood as frequent childhood mental conditions. The KMO sampling adequacy index was 0.72 in 2011 and 0.73 in 2015, while a cumulative variance percentage was of 34.0% in 2011 and 33.2% in 2015.

#### 3.1.2. Women Aged 15–44 Years

In women in this age group, the epidemiological factors identified in 2015 were already similar in 2011 but appearing less complex (Table 4). Mechanical pain factor was identified in 2011, factor not maintained during 2015, characterized by low back pain as the only condition and drugs such as opioids, muscle relaxants, NSAID. Other factors identified are comparable, such as the respiratory one, which includes asthma, allergic rhinitis, acute lower respiratory tract infection, but more pertaining drugs were recorded during 2015. The mental health factor was also comparable but appeared as a third factor in 2011 and as the first factor in 2015. Depression and anxiety were recorded during the mental health of 2011 with antidepressants, anxiolytics, and antiepileptics, while, during 2015, sleep, neurologic, and peripheral disorders were also recorded. The last factor identified was the endocrinological with iron deficiency in both years and hypothyroidism only in 2015. The KMO sampling adequacy index was 0.77 in 2011 and 0.74 in 2015 and a cumulative variance percentage was 47.0% in 2011 and 35.6% in 2015.

#### 3.1.3. Women Aged 45–65 Years

Scree plot identified the same four factors of 2015, in the same order but, generally, factors identified in 2011 were more complex in terms of clinical conditions number (Table 4). Anxiety, depression, sleep, neurologic, and peripheral disorders were recorded during 2011, while only anxiety, depression, and sleep disorder remained in the 2015 factor. Related drugs were comparable, as opioids remained presents for both years. The second factor identified was respiratory due to the presence of asthma and allergic rhinitis, with the addition of acute lower respiratory tract infection during 2011. This factor was mostly made up of related drugs such as antibiotics, adrenergics, decongestants, and corticosteroids. The third cardiometabolic factor was composed of diabetes, hypertension, obesity, disorders of lipid metabolism equally for both years, but more conditions appeared in 2011, such as glaucoma. The last factor identified for both was the osteometabolic, which was similarly made up of osteoporosis and calcium. The KMO index was 0.86 in 2011 and 0.80 in 2015, while the cumulative variance percentage was 55.0% in 2011 and 31.3% in 2015.

#### 3.1.4. Boys Aged 0–14 Years

This profile was similar to that observed for girls aged 0–14 years, both for 2011 and 2015 factors (Table 5). In fact, likewise, factors identified in 2015 in boys in this age group were generally already detected in 2011, but with the presence of an allergic/derma component. The same differences observed for girls in terms of conditions and factor were observed for boys. The KMO sampling adequacy index was 0.72 in 2011 and 0.74 in 2015, while the cumulative variance percentage was 34.0% in 2011 and 35.6% in 2015.

#### 3.1.5. Men Aged 15–44 Years

Among men of this age group, the order and the composition of epidemiological patterns identified in 2015 were not maintained in 2011 (Table 5). In fact, during 2011, four factors were identified. The first one, recognized as mental health/mechanical pain, was comparable with the first two identified during 2015. Moreover, this factor appeared without condition but was only made up of drugs such as opioids, antiepileptics, anxiolytics, and NSAID. During 2015, mental health and mechanical pain were split into two factors containing, in the first case, depression, substance use, anxiety, and sleep disorders with related drugs, and, in the second case, low back pain. Respiratory factor observed during 2015 was already present in 2011 both for disease and drugs. The last two 2011 factors identified as cardiometabolic and derma were not present in 2015. The KMO sampling adequacy index was 0.85 in 2011 and 0.75 in 2015, while the cumulative variance percentage of 26.0% in 2011 and 37.0% in 2015.

#### 3.1.6. Men Aged 45–65 Years

All three factors identified in 2015 were already present in 2011 but in a different order (Table 5). In this age group, respiratory factor was enriched with emphysema, chronic bronchitis, COPD for rather than other age groups, equally during 2015 and 2011. This factor appeared first during 2011 and last in 2015. The cardiometabolic factor seemed more complex in this age group for both years, due to the number of conditions that emerged such as hypertension, obesity, diabetes, ischemic heart disease, and gout. Mental health factor appeared firstly during 2015 and has become more complex than in 2011. The KMO sampling adequacy index was 0.82 in 2011 and 0.63 in 2015, while the cumulative variance percentage was 40.0% in 2011 and 30.4% in 2015.

## 4. Discussion

This study found that baseline epidemiologic patterns of multimorbidity and polypharmacy identified in the young and adult Spanish population during 2015 were already present in 2011 but with the addition of an allergic/derma pattern, which is not maintained in 2015. Globally, our findings also revealed that patterns identified in 2011 were more complex in terms of both disease and drugs; this could be a sign of an improvement and greater accuracy over the years in the computerized medical records systems. Other reason for the decreasing in the number of drugs taken by all age groups between 2011 and 2015 can be explained by the fact that after 2011, some medication was no longer reimbursed by the Spanish NHS, so this cannot be translated into a decrease in their use. We found that the complexity of patterns in terms of diseases and drugs, identified in both sexes, increases with age, and this trend remains unchanged in 2015.

The first difference identified can be represented by the presence of dermatitis and eczema as a condition more often diagnosed during 2011. In young subjects, the respiratory pattern was the most prevalent, even after four years. During 2015, the respiratory allergic component was predominant in children. This aspect was recorded during 2011, but it seems that respiratory conditions were better registered during 2015, as shown from the more accurate patterns resulted. Corroborating with our results, the high frequency of allergic and asthmatic components in childhood was widely discussed in the literature [15,16,17,18]. Similar was the case of childhood mental disorders and illnesses, conditions also found in 2011, with the addition in 2015 of the drugs for peptic ulcers and GERD, highlighting an increase in their use over the years attributable to prescriptive inappropriateness [19,20]. Additionally, a register of developmental and psychosocial disorders in children associated with antiepileptic treatments and attention deficit hyperactivity disorder (ADHD) treatments were established in both 2011 and 2015. The same pattern of drugs appeared in both sexes, but the diagnosis in girls seemed less accurate than in boys [21]. This could be explained as, in general, the clinic is more evident in boys, and among girls the symptoms are less intense, and therefore, a more general descriptor is used. For these reasons, since 2011, pediatricians started to collaborate with psychiatrists in the follow-up and treatment of affected children [22].

Various changes have been highlighted over the years among the age group 15–44 years in both sexes. Drugs such as cough suppressants and propulsives were dispensed to both men and women in 2011, also in younger and older age groups, but not in 2015, but this can be explained by the fact that after 2011, they were no longer reimbursed by the Spanish NHS. Another considerable difference is related to the mental factor that has become more complex in 2015, differently for men and women. Hence, during 2015, the mental factor was more prevalent among women. The prevalence of depression increased from 4.5% in 2011 to 6.7% in 2015, and more neurological disorders were diagnosed. This could be partly explained by an increase in psychophysical stress caused by more accelerated life rhythms over the years [23]. Similarly, in 2015, men were diagnosed with more disorders not present in 2011, and there was also evidence of substance use disorder, not present in women [24]. Substance use in men, in this age group, could be the cause of the worsening of the diagnosis picture in 2015; in fact, it appears to be a mechanical pain factor that was not present at all in 2011.

It is likely that as polypharmacy increases, drug dependence also increases, which leads to the development of a phenomenon of drug tolerance that complicates the overall clinical framework [25]. In women, it is noteworthy that the mental factor appeared in some psychosocial disorders, such as psychosocial disorders of childhood, combined with a drug cluster in which opioids appeared only in 2015. Perhaps this could be related to the higher prescription of tramadol in 2011, as this molecule was associated with the mechanical pattern. To date, several observational studies are alerting health authorities due to the adverse effects of opioid drugs associated with gabapentin. In fact, in Canada and France, there has been a warning about the risk of combining gabapentin and opioids, both in clinical practice and for recreational use [26,27]. In Ireland, the Medical Council has urged doctors to reduce the prescription of sedative drugs, including gabapentin [28]. Additionally, a recently published study linked the use of these drugs, especially pregabalin, to an increased risk of suicidal behavior, involuntary overdoses, injuries, traffic accidents, and crime [29]. Furthermore, among women, mechanical pain was detected in 2011 but not in 2015; in this year, the neurologic disorders that produce pain as neurologic disorders and peripheral neuropathy are included in the mental health patterns. A significant difference is, in fact, evident with men in the same age group, for whom, as in 2011, the mechanical pain factor remained in 2015.

Our results showed that in 2011 a cardiometabolic factor appeared in men in the 15–44 age group, while during 2015, in the older age group. It could be that until 2011, the occurrence of an episode of hypertension was sufficient to be diagnosed; however, with the subsequent establishment of new guidelines, the diagnosis has to be more accurate and well confirmed [30].

Furthermore, our findings also revealed that in 2011, as for 2015, the association between age and epidemiological pattern complexity is confirmed, as already discussed in literature [31,32]. Therefore, both for 2011 and 2015, among adults until 65 years, all the patterns appeared more complex than other age groups. In fact, the most predominant factors maintained over time were respiratory, cardiometabolic, and mental factors. Respiratory factor generally appeared more complex in 2011 than 2015, because it has been widely studied and identified the systematic association between asthma and allergic rhinitis; this has allowed for making a more accurate diagnosis [33,34,35]. Cardiometabolic factor appears similar for men and women with the addition of gout in men. This is in line with other studies, reporting that a prevalence rate of 1–2% for adults, underlining that it represents the most common inflammatory arthritis in men [36,37]. Another difference between sex was that this pattern in men included consequences of metabolic syndrome such as cardiovascular disease, ischemic heart disease, and cardiac arrhythmia, which is possibly due to increased cardiovascular risk in men, together with an increased incidence of ischemic heart and cerebrovascular diseases [38].

The mechanical pain in men aged 15–44 group in 2011 is included in the mental health pattern, while is separated in 2015. Contrarily, for women of the same age group, mechanical pain appeared only in 2011. The association of anxiety, depression, and somatic symptoms displayed in this pattern is well described, and somatic symptoms are mainly associated with emotional and brain functions, and they may reflect potential emotional conflicts that patients cannot face [39].

Finally, for the 45–56 age group, another gender difference can be highlighted, such as the presence of osteometabolic factor among women. This factor made up of osteoporosis and calcium, during 2011 also contained drugs affecting bone structure and mineralization that disappeared during 2015. The absence of these drugs in 2015 could be partly explained by the restrictions in use of bisphosphonates, recommended by the Spanish Agency of Medicines and Medical Devices in 2011, due to their association with a higher risk of atypical fractures [40].

In various patterns, we revealed potential DDIs, which could increase the risk of adverse health outcomes. Among them, we could highlight the use of inhaled beta-adrenergic agonists and corticosteroids, which decreased potassium levels, thus increasing the risk of arrhythmia [41]; the use of macrolides with inhaled beta-adrenergic and antihistamines, producing a QT prolongation and thus increasing the risk of arrhythmia [42]; the combined use of benzodiazepines and opioids, which increases sedation and respiratory depression [41].

### 4.1. Comparison with Other Studies

Multiple studies have been published in the recent years describing the different multimorbidity patterns, such as a study conducted in patients over 14 years old that described the existence of mechanical obesity, metabolic, neurovascular, liver disease, psychiatric substance abuse, anxiety, and depression-related patterns [8]. In addition, others studies only described the polypharmacy patterns [35]. However, in 2019, a study on multimorbidity and polypharmacy patterns showed the existence of some unexpected systematic associations among chronic diseases and drugs, as well as potential DDIs and prescribing cascades described in multimorbid patients [11]. Other authors had identified patterns between drugs and chronic disease in populations with a specific disease. For example, Hanlon et al. in 2018 describe the pattern and extent of multimorbidity and polypharmacy in patients with chronic obstructive pulmonary disease [43]. Nevertheless, our study described the patterns that influence to all the population. Aoki et al. in 2018 developed a study similar to ours identifying the multimorbidity patterns in a Japanese population, determining the effects on polypharmacy and dosage frequency [44].

The present study could be considered more exhaustive, because it compared the evolution of multimorbidity and polypharmacy patterns between 4 years in the same population, although this time span is not enough to detect long-term changes.

### 4.2. Strengths and Limitations

To our knowledge, this is the first large-scale population study comparing the differences observed in 4 years in the systematic associations among chronic diseases and dispensed drugs. The large population size of the EpiChron Cohort, together with the quality of data, resulting in reliable and representative results compared to those based only on medical records or drug use surveys [11]. In order to compare the same population at two different times, in this study, we have considered the population as an open cohort and, thus, not a cohort composed of a fixed number of members, but a dynamic cohort in which over time some subjects became lost and others are involved in the study. A population residing in a geographical area is, by definition, an open (or dynamic) cohort made up of individuals who contribute their personal time to the cohort, as long as they meet the membership criteria, i.e., place of residence, age, and health status. Therefore, having analyzed the variations in terms of multimorbidity and polypharmacy patterns in the population of Aragon, the cohort observed in 2011 and 2015 was considered as dynamic.

During the last five years, valuable information has been published regarding the security profile of numerous drugs, as was the case of benzodiazepines and opioids, allowing us to discuss our findings from both 2011 and 2015 in a more comprehensive manner. One of the essential methodological limitations of this study concerns the impossibility of including some drugs in the analyses due to multicollinearity with specific diseases, thus leading to the absence of specific drugs that would be, a priori, expected in some patterns. The issue of multicollinearity was also responsible for excluding the population aged >65 years from the analysis, which limited the comprehensiveness of the study. Nevertheless, in the present study, we used the same methodological criteria as the reference study to compare two populations that are as homogeneous as possible [11]. Furthermore, we conducted this study in order to assess the variations in most common clinical profiles among real-world population over the years. The 4 years evaluated were from 2011 to 2015 due to the availability of such data; in the future, a further survey may be carried out over more recent years. Providing information based on real-world data [45,46,47,48,49,50,51] may be a useful way to explore the dynamics in real clinical practice and to improve single-patient care model.

## 5. Conclusions

This study investigated the nature and complexity of a population, investigating the presence of systematic associations between diseases and drugs at two different times. We found that most clinical profiles were maintained over time as in the case of mental, cardiometabolic, mechanical, endocrinological, and osteometabolic patterns. Our findings revealed that baseline multimorbidity and polypharmacy patterns are maintained over time, as the nature of patterns identified in 2011 was also confirmed in 2015. Furthermore, our results also confirmed the existing association between age and clinical complexity, confirming a correlation between multimorbidity and ageing. The present study, therefore, confirmed systematic associations between diseases and drugs in the patterns over time. This could help in the early identification of potential interactions in multimorbid patients with a high risk of adverse health outcomes due to polypharmacy.

## Figures and Tables

**Table 1 ijerph-18-04422-t001:** Demographic and clinical characteristics of women in 2011 and 2015.

Subjects’ Characteristics	0–14 Years			15–44 Years			45–65 Years		
Women	2011	2015	*p* Value	2011	2015	*p* Value	2011	2015	*p* Value
**DEMOGRAPHIC**									
**Population** (N)	72,940	78,534		245,171	205,122		170,584	168,587	
**Age** (mean (SD))	7.79 (3.71)	7.03 (4.21)	<0.001	31.57 (8.21)	31.71 (8.45)	<0.001	54.23 (6.03)	54.43 (5.96)	<0.001
**Area of living** (n (%))			0.001 ^a^			<0.001 ^a^			<0.001 ^a^
**Urban**	43,911 (60.20%)	46,649 (59.40%)		155,773 (63.54%)	127,450 (62.13%)		109,249 (64.04%)	106,244 (63.02%)	
**Rural**	29,008 (39.77%)	31,885 (40.60%)		89,252 (36.40%)	77,672 (37.87%)		61,279 (35.92%)	62,343 (36.98%)	
**Unknown**	21 (0.03%)	-		146 (0.06%)	-		56 (0.03%)		
**Immigrant status** (n (%))			0.032 ^a^			<0.001 ^a^			<0.001 ^a^
**Native**	61,997 (85.00%)	67,740 (86.26%)		199,026 (81.18%)	168,839 (82.31%)		159,239 (93.35%)	156,311 (92.72%)	
**Immigrant**	10,168 (13.94%)	10,761 (13.70%)		46,100 (18.80%)	36,277 (17.69%)		11,331 (6.64%)	12,275 (7.28%)	
**Unknown**	775 (1.06%)	33 (0.04%)		45 (0.02%)	6 (0.00%)		14 (0.01%)	1 (0.00%)	
**Deprivation index** (n (%)) ^b^			<0.001 ^a^			<0.001 ^a^			0.007 ^a^
**Q**_1_	20,305 (27.84%)	22,448 (28.58%)		69,079 (28.18%)	55,733 (27.17%)		44,754 (26.24%)	43,546 (25.83%)	
**Q** _2_	18,719 (25.66%)	19,019 (24.22%)		60,847 (24.82%)	50,671 (24.70%)		43,587 (25.55%)	43,732 (25.94%)	
**Q** _3_	14,137 (19.38%)	15,556 (19.81%)		48,256 (19.68%)	41,415 (20.19%)		35,696 (20.93%)	35,040 (20.78%)	
**Q** _4_	19,743 (27.07%)	21,511 (27.39%)		66,913 (27.29%)	57,303 (27.94%)		46,512 (27.27%)	46,269 (27.45%)	
**Unknown**	36 (0.05%)	-		76 (0.03%)	-		35 (0.02%)	-	
**CLINICAL**									
**Number of chronic diseases ^e^**			<0.001			<0.001			<0.001
mean (SD)	0.67 (0.92)	1.00 (1.05)		0.89 (1.24)	1.47 (1.47)		2.28 (2.18)	3.06 (2.34)	
median (P_25_; P_75_)	0 (0; 1)	1 (0; 2)		0 (0; 1)	1 (0; 2)		2 (1; 3)	3 (1; 4)	
**Multimorbidity** (n (%)) ^c^	11,525 (15.80%)	20,022 (25.49%)	<0.001	56,798 (23.17%)	80,521 (39.26%)	<0.001	95,722 (56.11%)	120,101 (71.24%)	<0.001
**Number of drugs** ^d,e^			<0.001			<0.001			<0.001
mean (SD)	2.40 (2.42)	2.16 (2.09)		2.80 (3.12)	2.67 (2.71)		5.13 (4.66)	4.34 (3.75)	
median (P_25_; P_75_)	2 (0; 4)	2 (0; 3)		2 (0; 4)	2 (0; 4)		4 (1; 8)	4 (1; 6)	

^a^ Missing values were not considered when performing test and *p* value. ^b^ Deprivation index: degree of deprivation from the lowest (Q1) to the highest (Q4) of the administrative health area to which it belongs. ^c^ Defined as the coexistence of 2 or more chronic diseases. ^d^ Refers to different drugs dispensed at the third level of the anatomical, therapeutic, chemical (ATC) classification system. ^e^ Non-parametric test.

**Table 2 ijerph-18-04422-t002:** Demographic and clinical characteristics of men in 2011 and 2015.

Subjects’ Characteristics	0–14 Years			15–44 Years			45–65 Years		
Men	2011	2015	*p* Value	2011	2015	*p* Value	2011	2015	*p* Value
**DEMOGRAPHIC**									
**Population** (N)	77,391	82,893		260,915	190,658		173,389	161,778	
**Age** (mean (SD))	7.82 (3.72)	7.04 (4.21)	<0.001	31.68 (8.18)	31.54 (8.67)	0,768	54.00 (6.01)	54.36 (5.93)	<0.001
**Area of living** (n (%))			<0.001 ^a^			<0.001 ^a^			<0.001 ^a^
**Urban**	46,346 (59.89%)	48,943 (59.04%)		160,106 (61.36%)	113,262 (59.41%)		102,994 (59.40%)	94,223 (58.24%)	
**Rural**	31,022 (40.08%)	33,950 (40.96%)		100,728 (38.61%)	77,396 (40.59%)		70,349 (40.57%)	67,555 (41.76%)	
**Unknown**	23 (0.03%)	-		81 (0.03%)	-		46 (0.03%)		
**Immigrant status** (n (%))			<0.001 ^a^			<0.001 ^a^			<0.001 ^a^
**Native**	65,525 (84.67%)	71,506 (86.26%)		206,631 (79.19%)	160,073 (83.96%)		159,095 (91.76%)	149,258 (92.26%)	
**Immigrant**	11,040 (14.27%)	11,357 (13.70%)		54,219 (20.78%)	30,577 (16.04%)		14,292 (8.24%)	12,519 (7.74%)	
**Unknown**	826 (1.07%)	30 (0.04%)		65 (0.02%)	8 (0.00%)		2 (0.00%)	1 (0.00%)	
**Deprivation index** (n (%)) ^b^			<0.001 ^a^			<0.001 ^a^			0.011 ^a^
**Q** _1_	21,455 (27.72%)	23,695 (28.59%)		69,997 (26.83%)	49,759 (26.10%)		43,513 (25.10%)	40,042 (24.75%)	
**Q** _2_	19,695 (25.45%)	19,725 (23.80%)		64,037 (24.54%)	46,709 (24.50%)		44,237 (25.51%)	41,522 (25.67%)	
**Q** _3_	15,168 (19.60%)	16,465 (19.86%)		52,872 (20.26%)	39,962 (20.96%)		37,330 (21.53%)	34,511 (21.33%)	
**Q** _4_	21,052 (27.20%)	23,008 (27.76%)		73,953 (28.34%)	54,228 (28.44%)		48,285 (27.85%)	45,703 (28.25%)	
**Unknown**	21 (0.03%)	-		56 (0.02%)	-		24 (0.01%)	-	
**CLINICAL**									
**Number of chronic diseases ^e^**			<0.001			<0.001			<0.001
mean (SD)	0.76 (0.99)	1.12 (1.11)		0.62 (1.01)	1.14 (1.24)		1.70 (1.94)	2.48 (2.12)	
median (P_25_; P_75_)	0 (0; 1)	1 (0; 2)		0 (0; 1)	1 (0; 2)		1 (0; 3)	2 (1; 3)	
**Multimorbidity** (n (%)) ^c^	14,748 (19.06%)	24,386 (29.42%)	<0.001	38,788 (14.87%)	55,704 (29.22%)	<0.001	75,251 (43.40%)	99,176 (61.30%)	<0.001
**Number of drugs** ^d,e^			<0.001			<0.001			<0.001
mean (SD)	2.50 (2.50)	2.27 (2.20)		1.71 (2.34)	1.78 (2.10)		3.53 (3.88)	3.42 (3.32)	
median (P_25_; P_75_)	2 (0; 4)	2 (0; 3)		1 (0; 3)	1 (0; 3)		2 (0; 5)	3 (1; 5)	

^a^ Missing values were not considered when performing test and *p* value. ^b^ Deprivation index: degree of deprivation from the lowest (Q1) to the highest (Q4) of the administrative health area to which it belongs. ^c^ Defined as the coexistence of 2 or more chronic diseases. ^d^ Refers to different drugs dispensed at the third level of the anatomical, therapeutic, chemical (ATC) classification system. ^e^ Non-parametric test.

**Table 3 ijerph-18-04422-t003:** Comparison of multimorbidity and polypharmacy patterns identified in each age and sex group in 2011 and 2015.

Gender	0–14 Years		15–44 Years		45–65 Years	
	2011	2015	2011	2015	2011	2015
Women	Allergic–Derma	Respiratory–Acute Infection	Mechanical Pain	Mental Health	Mental Health	Mental Health
	Respiratory-Asthma–Acute Infection	Respiratory–Asthma–Allergic	Respiratory	Respiratory	Respiratory	Respiratory
	Allergic	Mental Health	Mental Health	Endocrinological	Cardiometabolic	Cardiometabolic
	Mental Health		Endocrinological		Osteometabolic	Osteometabolic
Men	Allergic–Derma	Respiratory–Acute Infection	Mental Health–Pain	Mental Health	Respiratory	Mental Health
	Respiratory–Asthma–Acute Infection	Respiratory–Asthma–Allergic	Respiratory–Allergic	Mechanical Pain	Cardiometabolic	Cardiometabolic
	Allergic	Mental Health	Cardiometabolic	Respiratory	Mental Health	Respiratory
	Mental Health		Derma			

**Table 4 ijerph-18-04422-t004:** Patterns of chronic diseases (EDC codes) and drugs (ATC codes) and factor loading scores in women. Diseases are highlighted in bold.

Year 2011		Prevalences	Values	Year 2015		Prevalences	Values
0–14 years							
**FACTOR 1: ALLERGIC/DERMA**		**Prev (%)**	**Values**	**FACTOR 1: RESPIRATORY/ACUTE INFECTION**		**Prev (%)**	**Values**
M01A	Anti-inflammatory and antirheumatic products, non-steroids	38.65	0.6462	H02A	Corticosteroids for systemic use, pain	9.40	0.6427
J01C	Beta-lactam antibacterials, penicillins	34.11	0.6454	RES02	Acute lower respiratory tract infection	11.06	0.6355
N02B	Other analgesics and antipyretics	17.85	0.5855	R03A	Adrenergics, inhalants	10.68	0.6224
R05C	Expectorants, excl. combinations with cough suppressants	22.57	0.5845	J01C	Beta-lactam antibacterials, penicillins	33.57	0.5882
R05D	Cough suppressants, excl, combinations with expectorants	22.14	0.5616	N02B	Other analgesics and antipyretics	22.57	0.5116
J01D	Other beta-lactam antibacterials	5.30	0.4862	J01F	Macrolides, lincosamides, and streptogramins	8.76	0.4816
S01A	Anti-infectives	6.86	0.4411	N05B	Anxiolytics	3.64	0.4570
J01F	Macrolides, lincosamides and streptogramins	8.24	0.4225	S01A	Anti-infectives	9.63	0.4271
D07A	Corticosteroids, plain	7.87	0.4198	M01A	Anti-inflammatory and antirheumatic products, non-steroids	34.45	0.4174
A03F	Propulsives	2.04	0.3905	D07A	Corticosteroids, plain	8.05	0.4097
D01A	Antifungals for topical use	3.44	0.3817	D01A	Antifungals for topical use	3.97	0.3684
N05B	Anxiolytics	3.71	0.3750	A07C	Electrolytes with carbohydrates	4.15	0.3648
D06A	Antibiotics for topical use	4.34	0.3681	D06A	Antibiotics for topical use	5.07	0.3583
SKN02	Dermatitis and eczema	18.13	0.2929				
**FACTOR 2: RESPIRATORY/ASTHMA/** **ACUTE INFECTION**		**Prev (%)**	**Values**	**FACTOR 2: RESPIRATORY/ASTHMA/** **ALLERGIC**		**Prev (%)**	**Values**
H02A	Corticosteroids for systemic use, plain	6.93	0.4682	R06A	Antihistamines for systemic use	13.63	0.6105
R03A	Adrenergics, inhalants	7.50	0.8946	ALL03	Allergic rhinitis	4.23	0.7546
RES02	Acute lower respiratory tract infection	8.21	0.7506	S01G	Decongestants and antiallergics	2.68	0.7419
ASMA	Asthma	6.25	0.6038	R01A	Decongestants and other nasal preparations for topical use	3.90	0.6744
				ASMA	Asthma	7.18	0.3489
**FACTOR 3: ALLERGIC**		**Prev (%)**	**Values**				
R06A	Antihistamines for systemic use	10.50	0.5823				
ALL03	Allergic rhinitis	2.90	0.8316				
S01G	Decongestants and antiallergics	1.92	0.7065				
R01A	Decongestants and other nasal preparations for topical use	3.03	0.6528				
**FACTOR 4: MENTAL HEALTH**		**Prev (%)**	**Values**	**FACTOR 3: MENTAL HEALTH**		**Prev (%)**	**Values**
N06B	Psychostimulants, agents used for ADHD and nootropics	0.89	0.7123	N03A	Antiepileptics	0.36	0.6693
N03A	Antiepileptics	0.36	0.6379	N06B	Psychostimulants, agents used for ADHD and nootropics	0.74	0.5403
NUR19	Developmental disorder	1.19	0.6150	NUR19	Developmental disorder	2.15	0.3793
PSY14	Psychosocial disorders of childhood	3.40	0.3113	A02B	Drugs for peptic ulcers and GERD	0.69	0.3761
				PSY14	Psychosocial disorders of childhood	5.36	0.3287
**15–44 years**							
**FACTOR 1: MECHANICAL PAIN**		**Prev (%)**	**Values**				
M01A	Anti-inflammatory and antirheumatic products, non-steroids	30.97	0.7664				
M03B	Muscle relaxants, centrally acting agents	4.08	0.5416				
A02B	Drugs for peptic ulcer and GERD	10.67	0.5046				
N02B	Other analgesics and antipyretics	19.65	0.5007				
M02A	Topical products for joint and muscular pain	3.80	0.4578				
N02A	Opioids	2.68	0.4304				
J01C	Beta-lactam antibacterials, penicillins	19.96	0.3998				
MUS14	Low back pain	4.20	0.3607				
R05D	Cough suppressants, excl. combinations with expectorants	9.30	0.3497				
A03F	Propulsives	4.27	0.3157				
**FACTOR 2: RESPIRATORY**		**Prev (%)**	**Values**	**FACTOR 1: RESPIRATORY**		**Prev (%)**	**Values**
R05C	Expectorants, excl. combinations with cough suppressants	14.62	0.4734	M01A	Anti-inflammatory and antirheumatic products, non-steroids	30.85	0.3224
J01F	Macrolides. lincosamides and streptogramins	7.94	0.3563	R06A	Antihistamines for systemic use	14.83	0.8167
S01C	Anti-inflammatory agents and anti-infectives in combination	2.15	0.9123	R03A	Adrenergics, inhalants	5.24	0.7087
R03A	Adrenergics, inhalants	3.95	0.8991	R01A	Decongestants and other nasal reparations for topical use	8.50	0.6800
ASMA	Asthma	4.15	0.6915	S01G	Decongestants and antiallergics	3.10	0.6329
R06A	Antihistamines for systemic use	11.12	0.6647	ASMA	Asthma	6.67	0.4935
R01A	Decongestants and other nasal preparations for topical use	6.37	0.5650	RES02	Acute lower respiratory tract infection	2.38	0.4617
RES02	Acute lower respiratory tract infection	2.25	0.5564	ALL03	Allergic rhinitis	12.64	0.4243
ALL03	Allergic rhinitis	7.27	0.3956	H02A	Corticosteroids for systemic use. plain	3.30	0.4065
H02A	Corticosteroids for systemic use, plain	2.35	0.3574	J01F	Macrolides, lincosamides and streptogramins	9.55	0.3837
				J01C	Beta-lactam antibacterials, penicillins	21.02	0.3651
				J01M	Quinolone antibacterials	3.64	0.3413
				J01D	Other beta-lactam antibacterials	3.42	0.3320
				N02B	Other analgesics and antipyretics	20.89	0.3169
				D07A	Corticosteroids, plain	5.54	0.3086
**FACTOR 3: MENTAL HEALTH**		**Prev (%)**	**Values**	**FACTOR 2: MENTAL HEALTH**		**Prev (%)**	**Values**
N06A	Antidepressants	6.10	0.9314	N06A	Antidepressants	6.95	0.8600
N05B	Anxiolytics	8.86	0.7156	N03A	Antiepileptics	2.75	0.7610
N03A	Antiepileptics	2.22	0.6426	N05B	Anxiolytics	11.11	0.7584
PSY09	Depression	4.55	0.6301	N05A	Antipsychotics	2.03	0.5738
N05A	Antipsychotics	1.83	0.5151	PSY09	Depression	6.76	0.5535
PSY01	Anxiety, neuroses	2.65	0.4704	A02B	Drugs for peptic ulcers and GERD	10.42	0.4688
N02C	Antimigraine preparations	1.48	0.2683	N02A	Opioids	3.83	0.4575
				PSY01	Anxiety, neuroses	4.89	0.4333
				PSY19	Sleep disorders of nonorganic origin	3.65	0.3776
				N02C	Antimigraine preparations	1.74	0.3742
				NUR21	Neurologic disorders, other	2.33	0.3556
				NUR03	Peripheral neuropathy, neuritis	2.60	0.3093
**FACTOR 4: ENDOCRINOLOGICAL**		**Prev (%)**	**Values**	**FACTOR 3: ENDOCRINOLOGICAL**		**Prev (%)**	**Values**
B03A	Iron preparations	7.73	0.9181	B03A	Iron preparations	8.97	0.7959
H03C	Iodine therapy	4.24	0.7731	H03C	Iodine therapy	5.61	0.6469
HEM02	Iron deficiency, other deficiency anemias	4.10	0.5908	HEM02	Iron deficiency, other deficiency anemias	6.18	0.5369
B03B	Vitamin B12 and folic acid	3.53	0.5032	B03B	Vitamin B12 and folic acid	3.99	0.4798
G03A	Hormonal contraceptives for systemic use	3.03	0.3399	H03A	Thyroid preparations	4.30	0.4306
C05C	Capillary stabilizing agents	3.02	0.2817	END04	Hypothyroidism	6.29	0.3658
**45–65 years**							
**FACTOR 1: MENTAL HEALTH**		**Prev (%)**	**Values**	**FACTOR 1: MENTAL HEALTH**		**Prev (%)**	**Values**
N06A	Antidepressants	16.63	0.8254	N06A	Antidepressants	18.21	0.8980
N05B	Anxiolytics	22.35	0.7021	N05B	Anxiolytics	24.67	0.6682
N03A	Antiepileptics	5.70	0.5976	PSY09	Depression	16.81	0.6131
N05C	Hypnotics and sedatives	6.12	0.5944	N05C	Hypnotics and sedatives	6.08	0.5592
PSY09	Depression	12.93	0.5871	N03A	Antiepileptics	7.01	0.5406
N02A	Opioids	8.24	0.4676	PSY01	Anxiety, neuroses	6.70	0.4116
A02B	Drugs for peptic ulcer and GERD	30.90	0.4483	N02A	Opioids	10.24	0.3805
PSY01	Anxiety, neuroses	4.13	0.4187	PSY19	Sleep disorders of nonorganic origin	10.29	0.3618
PSY19	Sleep disorders of nonorganic origin	6.54	0.4111	A02B	Drugs for peptic ulcers and GERD	29.06	0.3379
M01A	Anti-inflammatory and antirheumatic products, non-steroids	46.56	0.4095				
A03F	Propulsives	6.28	0.3674				
M03B	Muscle relaxants, centrally acting agents	7.11	0.3614				
MUS13	Cervical pain syndromes	2.38	0.3128				
MUS14	Low back pain	7.52	0.2733				
NUR21	Neurologic disorders, other	3.57	0.2617				
NUR03	Peripheral neuropathy, neuritis	4.59	0.2524				
**FACTOR 2: RESPIRATORY**		**Prev (%)**	**Values**	**FACTOR 2: RESPIRATORY**		**Prev (%)**	**Values**
N02B	Other analgesics and antipyretics	30.15	0.3050	R03A	Adrenergics, inhalants	7.93	0.7548
R03A	Adrenergics, inhalants	6.21	0.8711	R06A	Antihistamines for systemic use	16.74	0.7487
R05C	Expectorants, excl. combinations with cough suppressants	20.41	0.7092	R01A	Decongestants and other nasal preparations for topical use	8.22	0.6301
RES02	Acute lower respiratory tract infection	4.46	0.7032	ASMA	Asthma	6.38	0.5872
R06A	Antihistamines for systemic use	13.51	0.6205	H02A	Corticosteroids for systemic use, pain	6.77	0.4867
ASMA	Asthma	4.45	0.5862	J01F	Macrolides, lincosamides, and streptogramins	10.82	0.4468
R01A	Decongestants and other nasal preparations for topical use	7.03	0.5761	J01M	Quinolone antibacterials	6.61	0.4313
J01F	Macrolides, lincosamides, and streptogramins	9.29	0.5400	ALL03	Allergic rhinitis	10.50	0.4032
J01M	Quinolone antibacterials	6.42	0.5128	J01C	Beta-lactam antibacterials, penicillins	17.97	0.3853
H02A	Corticosteroids for systemic use, plain	5.25	0.5007	N02B	Other analgesics and antipyretics	29.09	0.3269
J01C	Beta-lactam antibacterials, penicillins	17.98	0.4622				
R05D	Cough suppressants, excl. combinations with expectorants	11.74	0.4230				
ALL03	Allergic rhinitis	6.33	0.2741				
**FACTOR 3: CARDIOMETABOLIC**		**Prev (%)**	**Values**	**FACTOR 3: CARDIOMETABOLIC**		**Prev (%)**	**Values**
DIAB	Diabetes	4.99	0.7288	HTA	Hypertension	20.49	0.9601
HTA	Hypertension	19.06	0.6791	C09A	ACE inhibitors, plain	5.06	0.7041
NUT03	Obesity	9.00	0.6258	DIAB	Diabetes	5.58	0.5854
B01A	Antithrombotic agents	6.49	0.4258	NUT03	Obesity	11.62	0.5014
CAR11	Disorders of lipid metabolism	23.70	0.3817	B01A	Antithrombotic agents	6.51	0.3699
ARTRITIS	Degenerative joint disease	11.66	0.3318	CAR11	Disorders of lipid metabolism	32.89	0.2951
C05C	Capillary stabilizing agents	9.78	0.2882				
EYE08	Glaucoma	2.93	0.2823				
GSU08	Varicose veins of lower extremities	15.78	0.2771				
**FACTOR 4: OSTEOMETABOLIC**		**Prev (%)**	**Values**	**FACTOR 4: OSTEOMETABOLIC**		**Prev (%)**	**Values**
M05B	Drugs affecting bone structure and mineralization	6.29	0.9690	A12A	Calcium	6.10	0.8032
A12A	Calcium	8.96	0.8944	END02	Osteoporosis	8.98	0.7869
END02	Osteoporosis	9.45	0.8609				

Abbreviations: ATC, anatomical therapeutic chemical classification; COPD, chronic obstructive pulmonary disease; EDC, expanded diagnostic clusters; GERD, gastro-esophageal reflux disease; Prev, prevalenc.

**Table 5 ijerph-18-04422-t005:** Patterns of chronic diseases (EDC codes) and drugs (ATC codes) and factor loading scores in men. Diseases are highlighted in bold.

Year 2011				Year 2015			
0–14 years							
**FACTOR 1: ALLERGIC/DERMA**		**Prev** **(%)**	**Values**	**FACTOR 1: RESPIRATORY/ACUTE INFECTION**		**Prev** **(%)**	**Values**
J01C	Beta-lactam antibacterials, penicillins	34.08	0.6579	H02A	Corticosteroids for systemic use, pain	11.77	0.6877
M01A	Anti-inflammatory and antirheumatic products, non-steroids	38.23	0.6372	RES02	Acute lower respiratory tract infection	13.69	0.6748
N02B	Other analgesics and antipyretics	17.98	0.6097	R03A	Adrenergics, inhalants	13.79	0.6683
R05C	Expectorants, excl. combinations with cough suppressants	22.80	0.5800	J01C	Beta-lactam antibacterials, penicillins	33.48	0.5854
R05D	Cough suppressants, excl. combinations with expectorants	22.40	0.5611	R03B	Other drugs for obstructive airway diseases, inhalants	4.05	0.5520
J01D	Other beta-lactam antibacterials	4.75	0.4832	N02B	Other analgesics and antipyretics	22.76	0.5332
S01A	Anti-infectives	6.97	0.4410	J01F	Macrolides, lincosamides, and streptogramins	8.83	0.5120
A07C	Electrolytes with carbohydrates	3.12	0.4302	N05B	Anxiolytics	3.49	0.4556
D07A	Corticosteroids, plain	9.50	0.4195	S01A	Anti-infective	9.79	0.4545
J01F	Macrolides, lincosamides, and streptogramins	8.42	0.4027	D07A	Corticosteroids, plain	9.67	0.4018
N05B	Anxiolytics	3.56	0.4009	M01A	Anti-inflammatory and antirheumatic products, non-steroids	34.58	0.3990
A03F	Propulsives	1.91	0.3946	A07C	Electrolytes with carbohydrates	4.48	0.3666
D06A	Antibiotics for topical use	5.00	0.3710	D01A	Antifungals for topical use	3.36	0.3452
H02A	Corticosteroids for systemic use, plain	8.80	0.3262	D06A	Antibiotics for topical use	5.64	0.3344
SKN02	Dermatitis and eczema	16.84	0.2812				
**FACTOR 2: RESPIRATORY/ASTHMA/ACUTE INFECTION**		**Prev** **(%)**	**Values**	**FACTOR 2: RESPIRATORY/ASTHMA/** **ALLERGIC**		**Prev** **(%)**	**Values**
R03A	Adrenergics, inhalants	10.09	0.9215	R06A	Antihistamines for systemic use	14.54	0.6159
R03B	Other drugs for obstructive airway diseases, inhalants	3.59	0.8434	ALL03	Allergic rhinitis	5.34	0.7213
RES02	Acute lower respiratory tract infection	10.08	0.7459	S01G	Decongestants and antiallergics	3.86	0.6773
ASMA	Asthma	9.28	0.6818	R01A	Decongestants and other nasal preparations for topical use	4.33	0.6734
				ASMA	Asthma	10.96	0.4222
**FACTOR 3: ALLERGIC**		**Prev** **(%)**	**Values**				
R06A	Antihistamines for systemic use	11.26	0.6047				
ALL03	Allergic rhinitis	3.77	0.7499				
R01A	Decongestants and other nasal preparations for topical use	3.40	0.7494				
S01G	Decongestants and antiallergics	2.94	0.6728				
**FACTOR 4: MENTAL HEALTH**		**Prev** **(%)**	**Values**	**FACTOR 3: MENTAL HEALTH**		**Prev** **(%)**	**Values**
N06B	Psychostimulants, agents used for ADHD and nootropics	2.46	0.9564	N06B	Psychostimulants, agents used for ADHD and nootropics	2.18	0.7213
PSY05	Attention deficit disorder	1.97	0.8148	N03A	Antiepileptics	0.39	0.6562
NUR19	Developmental disorder	2.10	0.3823	PSY05	Attention deficit disorder	1.92	0.5889
PSY14	Psychosocial disorders of childhood	5.70	0.3139	PSY14	Psychosocial disorders of childhood	8.59	0.3968
				NUR19	Developmental disorder	3.89	0.3857
				A02B	Drugs for peptic ulcers and GERD	0.60	0.3324
**15–44 years**							
**FACTOR 1: MENTAL HEALTH/MECHANICAL PAIN**		**Prev** **(%)**	**Values**	**FACTOR 1: MENTAL HEALTH**		**Prev** **(%)**	**Values**
N03A	Antiepileptics	1.67	0.7159	N06A	Antidepressants	3.74	0.8979
N05C	Hypnotics and sedatives	0.97	0.6100	N05C	Hypnotics and sedatives	1.10	0.7614
N05B	Anxiolytics	4.60	0.6036	N05A	Antipsychotics	2.00	0.7482
N05A	Antipsychotics	1.53	0.5564	N05B	Anxiolytics	6.92	0.6522
A02B	Drugs for peptic ulcer and GERD	7.94	0.5129	N03A	Antiepileptics	2.45	0.6442
N02A	Opioids	1.90	0.5024	PSY09	Depression	3.47	0.6005
M01A	Anti-inflammatory and antirheumatic products, non-steroids	23.05	0.4126	PSY02	Substance use	2.79	0.4973
N02B	Other analgesics and antipyretics	14.24	0.3849	PSY01	Anxiety neuroses	2.55	0.4801
				PSY19	Sleep disorders of nonorganic origin	3.12	0.4604
				**FACTOR 2: MECHANICAL PAIN**		**Prev** **(%)**	**Values**
				M01A	Anti-inflammatory and antirheumatic products, non-steroids	25.68	0.7741
				N02B	Other analgesics and antipyretics	17.05	0.6115
				A02B	Drugs for peptic ulcers and GERD	8.49	0.5996
				J01C	Beta-lactam antibacterials, penicillins	18.01	0.5105
				N02A	Opioids	2.92	0.4920
				MUS14	Low back pain	4.18	0.4663
				H02A	Corticosteroids for systemic use, pain	2.58	0.4642
				J01F	Macrolides, lincosamides, and streptogramins	7.10	0.4037
				B01A	Antithrombotic agents	2.01	0.3980
**FACTOR 2: RESPIRATORY**		**Prev** **(%)**	**Values**	**FACTOR 3: RESPIRATORY**		**Prev** **(%)**	**Values**
H02A	Corticosteroids for systemic use, plain	1.65	0.3883	RES02	Acute lower respiratory tract infection	2.05	0.3838
R01A	Decongestants and other nasal preparations for topical use	4.73	0.7461	R03A	Adrenergics, inhalants	4.54	0.7900
R06A	Antihistamines for systemic use	7.83	0.6764	R06A	Antihistamines for systemic use	11.97	0.7005
R05C	Expectorants, excl. combinations with cough suppressants	10.13	0.5909	ASMA	Asthma	6.89	0.6227
ALL03	Allergic rhinitis	6.06	0.5124	R01A	Decongestants and other nasal preparations for topical use	6.99	0.5562
J01F	Macrolides, lincosamides, and streptogramins	5.22	0.4957	ALL03	Allergic rhinitis	12.12	0.4093
ASMA	Asthma	3.53	0.4573				
R05D	Cough suppressants, excl. combinations with expectorants	5.99	0.4383				
J01C	Beta-lactam antibacterials, penicillins	15.47	0.3796				
**FACTOR 3: CARDIOMETABOLIC**		**Prev** **(%)**	**Values**				
HTA	Hypertension	2.05	0.7494				
NUT03	Obesity	2.62	0.5822				
CAR11	Disorders of lipid metabolism	6.10	0.5101				
B01A	Antithrombotic agents	1.61	0.5063				
**FACTOR 4: DERMA**		**Prev** **(%)**	**Values**				
SKN02	Dermatitis and eczema	4.94	0.8586				
D07A	Corticosteroids, plain	3.62	0.6772				
D01A	Antifungals for topical use	3.15	0.4985				
**45–65 years**							
**FACTOR 1: RESPIRATORY**		**Prev** **(%)**	**Values**	**FACTOR 3: RESPIRATORY**		**Prev** **(%)**	**Values**
R05C	Expectorants, excl. combinations with cough suppressants	14.43	0.7394	N02B	Other analgesics and antipyretics	22.35	0.3056
R06A	Antihistamines for systemic use	8.06	0.6970	RES04	Emphysema, chronic bronchitis, COPD	3.64	0.3491
R01A	Decongestants and other nasal preparations for topical use	5.05	0.6485	R03A	Adrenergics, inhalants	5.88	0.8130
RES02	Acute lower respiratory tract infection	3.15	0.5805	R06A	Antihistamines for systemic use	11.10	0.7063
J01F	Macrolides, lincosamides, and streptogramins	5.57	0.5787	RES02	Acute lower respiratory tract infection	3.45	0.5897
R05D	Cough suppressants, excl. combinations with expectorants	7.00	0.5409	R01A	Decongestants and other nasal preparations for topical use	6.26	0.5803
J01C	Beta-lactam antibacterials, penicillins	14.33	0.5140	ASMA	Asthma	3.43	0.5666
N02B	Other analgesics and antipyretics	21.11	0.5065	J01M	Quinolone antibacterials	5.53	0.4548
M01A	Anti-inflammatory and antirheumatic products, non-steroids	32.37	0.4865	J01F	Macrolides, lincosamides, and streptogramins	6.91	0.4383
J01M	Quinolone antibacterials		0.4676	J01C	Beta-lactam antibacterials, penicillins	15.46	0.3981
ALL03	Allergic rhinitis	4.03	0.4222	ALL03	Allergic rhinitis	7.53	0.3589
ASMA	Asthma	2.05	0.4190				
D07A	Corticosteroids, plain	32.37	0.3434				
M02A	Topical products for joint and muscular pain	6.25	0.3159				
RES04	Emphysema, chronic bronchitis. COPD	2.69	0.2818				
**FACTOR 2: CARDIOMETABOLIC**		**Prev** **(%)**	**Values**	**FACTOR 2: CARDIOMETABOLIC**		**Prev** **(%)**	**Values**
A02B	Drugs for peptic ulcer and GERD	24.46	0.3434	A02B	Drugs for peptic ulcer and gastro-esophageal reflux disease (gord)	25.26	0.3952
HTA	Hypertension	22.12	0.8007	B01A	Antithrombotic agents	11.01	0.7832
B01A	Antithrombotic agents	9.93	0.6619	HTA	Hypertension	27.96	0.6610
DIAB	Diabetes	8.73	0.6547	IHD	Ischemic heart disease	4.07	0.6085
C09C	Angiotensin II receptor blockers (ARBs), plain	6.49	0.6080	DIAB	Diabetes	11.00	0.5750
IHD	Ischemic heart disease	3.23	0.5763	C09C	Angiotensin II receptor blockers (ARBs), plain	6.85	0.5396
NUT03	Obesity	6.73	0.5377	CAR16	Cardiovascular disorders, other	2.14	0.4854
CAR11	Disorders of lipid metabolism	26.32	0.4817	CAR09	Cardiac arrhythmia	2.50	0.4723
RHU02	Gout	2.88	0.3703	NUT03	Obesity	10.24	0.4283
				CAR11	Disorders of lipid metabolism	39.37	0.3296
				RHU02	Gout	4.17	0.3014
**FACTOR 3: MENTAL HEALTH**		**Prev** **(%)**	**Values**	**FACTOR 1: MENTAL HEALTH**		**Prev** **(%)**	**Values**
N06A	Antidepressants	6.04	0.9434	N06A	Antidepressants	7.22	0.7887
PSY09	Depression	4.42	0.7844	N05B	Anxiolytics	12.79	0.7326
N05B	Anxiolytics	10.25	0.7607	N03A	Antiepileptics	5.10	0.6613
PSY19	Sleep disorders of nonorganic origin	3.60	0.3751	PSY09	Depression	6.86	0.5530
PSY02	Substance use	2.61	0.3104	N02A	Opioids	6.60	0.4891
				PSY01	Anxiety, neuroses	3.04	0.4447
				M01A	Anti-inflammatory and antirheumatic products, non-steroids	31.42	0.4166
				PSY19	Sleep disorders of nonorganic origin	6.66	0.3594
				MUS14	Low back pain	6.13	0.3367
				MUS13	Cervical pain syndromes	2.48	0.3161
				NUR21	Neurologic disorders, other	3.69	0.2959

**Abbreviations:** ATC, anatomical therapeutic chemical classification; COPD, chronic obstructive pulmonary disease; EDC, expanded diagnostic clusters; GERD, gastro-esophageal reflux disease; Prev, Prevalence.

## Data Availability

The data used in this study cannot be publicly shared, because of restrictions imposed by the Aragon Health Sciences Institute (IACS) and asserted by the Clinical Research Ethics Committee of Aragon (CEICA, ceica@aragon.es). The authors who accessed the data belong to the EpiChron Research Group of IACS, and received permission from IACS to utilize the data for this specific study, thus implying its exclusive use by the researchers appearing in the project protocol approved by CEICA.

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
