# Peer review of "Changes in Multimorbidity and Polypharmacy Patterns in Young and Adult Population over a 4-Year Period: A 2011–2015 Comparison Using Real-World Data"

_ijerph, 2021, doi:10.3390/ijerph18094422_

Round 1
Reviewer 1 Report
I thank the authors and the editor for the revised version of the manuscript "Changes of multimorbidity and polypharmacy patterns in young and adult population over a 4-year period: a 2011-2015 comparison using real-world data."
I think the manuscript is very interesting. Nevertheless, I think there are some points to be improved, especially in the
- It is not clear what proportion of individuals are present in 2011 and 2015 and how this could affect results.
- In line with the previous point, I wondered why the authors choose to perform an unpaired t-test, since to my understanding, they test for differences between the two years under study by age group and sex.
- There is a lack of description of the deprivation index. How was it defined?
- The number of drugs is it composed of all the drugs used in the year of the survey? It is not clear the time of the assessment of the variable.
- Lines 138-139: If the cohorts are composed essentially of the same subjects, it is pretty predictable that the age increase and drug use increase with age.
- Lines 247-252: the similarities between 2011 and 2015 are pretty predictable since they are composed essentially by the same persons.
- Lines 279-280: It seems to be a missing verb.
- Lines 281-282: could be the increase be significant in 4 years?
- Line 350: in the phrase "Besides, in 2019, a study 349 on multimorbidity and polypharmacy patterns and shows the existence" is not precise, maybe they could have used "and shows" but they should have used "had shown".
- In the paragraph on strength and limitations, the authors should explain the concept of open cohort and how it applies to their study.
- A few references are a little dated, or better reference exist:
- Reference 1: A recent systematic review on polypharmacy (Sirois et al. Polypharmacy Definitions for Multimorbid Older Adults Need Stronger Foundations to Guide Research, Clinical Practice and Public Health. Pharmacy (Basel). 2019;7(3). 2019
- Reference number 5: there is no review yet, but more recent studies, as Chang et al. Polypharmacy, hospitalization, and mortality risk: a nationwide cohort study. Sci. Rep. 10(1) 2020 or Schöttker et al. Updated analysis on polypharmacy and mortality from the ESTHER study. Eur. J. Clin. Pharmacol. 74(7): 981-982 2018.
- Reference number 14 refers to a Taiwan study, not comparable to the population of Spain
Author Response
Response to Reviewer 1 Comments
Point 1: I thank the authors and the editor for the revised version of the manuscript "Changes of multimorbidity and polypharmacy patterns in young and adult population over a 4-year period: a 2011-2015 comparison using real-world data." I think the manuscript is very interesting. Nevertheless, I think there are some points to be improved, especially in the
Response 1: Dear Reviewer, on behalf of all the authors, we really appreciate your interest in our work.
Point 2: It is not clear what proportion of individuals are present in 2011 and 2015 and how this could affect results.
Response 2: As stated in Lines 149-151 on Page 4, subjects up to 65 years old identified in the Aragon region were 1,000,390 during 2011 and 887,572 during 2015. Also, in Lines 388-391 on Page 23, we specified that these two proportions of subjects are different as we decided to use an open cohort: “In order to compare the same population at two different times, in this study we have considered the population as an open cohort and thus not a cohort composed of a fixed number of members, but a dynamic cohort in which over time some subjects get lost and others are involved in the study.”
We analysed in each year all the individuals users of the public health system in Aragon and aged up to 65 years.
Point 3: In line with the previous point, I wondered why the authors choose to perform an unpaired t-test, since to my understanding, they test for differences between the two years under study by age group and sex.
Response 3: Dear Reviewer, because the primary purpose of our work was to compare and analyse the change in patterns of multimorbidity and polypharmacy in the same population, identified as an open cohort, we first compared how the same population varied over the 4 years. This comparison was performed using the chi-squared test for categorical variables or the unpaired t-test for numerical variables, as appropriate, so comparing changing in characteristics such as: age, area of living, immigrant status, deprivation index, number of chronic diseases, multimorbidity, number of drugs. Please, see Table 1 and Table 2 referred respectively to women and men.
Point 4: There is a lack of description of the deprivation index. How was it defined?
Response 4: Thanks for your advice. We’ve added a description of the deprivation index in the ‘Statistical analyses’ subsection in Lines 104-106 on Page 3: “The deprivation index is strictly related to the census section of subjects which represents the degree of deprivation from the lowest (Q1) to the highest (Q4) of the administrative health area to which it belongs”.
The construction of the deprivation index was published elsewhere:
Dea, M.L.C.; Bellido, E.O.; Solana, C.F.; Palacio, I.A.; Del Hombrebueno, G.G.-C.R.; Sancho, B.A. Construction of a deprivation index by Basic Healthcare Area in Aragon using Population and Housing Census 2011. Rev. Esp. Salud Publica 2018, 92, e201812087.
Point 5: The number of drugs is it composed of all the drugs used in the year of the survey? It is not clear the time of the assessment of the variable.
Response 5: Yes, the number of drugs and diseases are referred to the specific reference year (2011 and 2015). We’ve provided to specify it in the ‘Statistical analyses’ subsection in Lines 101-104 on Page 3: “Patients’ characteristics compared were age, area of living, immigrant status, deprivation index and number of chronic diseases, multimorbidity, number of drugs dispensed during the reference year”.
Point 6: Lines 138-139: If the cohorts are composed essentially of the same subjects, it is pretty predictable that the age increase and drug use increase with age.
Response 6: Thanks for the input. With this sentence we mean that the first results detected was that the trend of the statistically significant correlation between age and complexity was confirmed after 4 years. We rephrased the sentence in Lines 152-154 on Page 4: “Firstly, for both years 2011 and 2015, we detected a statistically significant increase in the number of drugs and chronic conditions for both sexes as age increases”.
Point 7: Lines 247-252: the similarities between 2011 and 2015 are pretty predictable since they are composed essentially by the same persons.
Response 7: We perfectly understand your concern, but we conducted a study assessing the changes and nature of patterns of multimorbidity and polypharmacy in the same population over 4 years. So, we certainly revealed similarities since much of the population of 2011 coincides with that of 2015, but we analysed in detail the differences in the number of drugs prescribed and in the type of diseases diagnosed. For example, we noticed how in some cases, certain clinical framework that were more complex in 2011, seem less complex in 2015 because they are better diagnosed, or moreover, how some trends of inappropriate prescribing are maintained over the years despite being the cause of long-term negative outcomes.
Point 8: Lines 279-280: It seems to be a missing verb.
Response 8: Thanks, we added the missing verb in the sentence: “Hence, during 2015, the mental factor was more prevalent among women”.
Point 9: Lines 281-282: could be the increase be significant in 4 years?
Response 9: We certainly agree that a time span of four years cannot detect long-term changes, which is why we hope to repeat this study by analysing even longer periods. Nevertheless, we have stated that the complexity of the mental factor can be partially explained due to the impact of Everyday Stressors which in 4 years could be one of the reasons for the increase or worsening of psychosocial complications.
Point 10: Line 350: in the phrase "Besides, in 2019, a study 349 on multimorbidity and polypharmacy patterns and shows the existence" is not precise, maybe they could have used "and shows" but they should have used "had shown".
Response 10: Thanks, we corrected with “had shown”.
Point 11: In the paragraph on strength and limitations, the authors should explain the concept of open cohort and how it applies to their study.
Response 11: Thanks for the advice. We added some statements regarding the open cohort in Lines 391-396 on Page 23: “A population residing in a geographical area is, by definition, an open (or dynamic) cohort made up of individuals who contribute their person-time to the cohort, as long as they meet the membership criteria, i.e., place of residence, age and health status. Therefore, having analysed the variations in terms of multimorbidity and polypharmacy patterns in the population of Aragon, the cohort observed in 2011 and 2015 was considered as dynamic”.
Point 12: A few references are a little dated, or better reference exist: Reference 1: A recent systematic review on polypharmacy (Sirois et al. Polypharmacy Definitions for Multimorbid Older Adults Need Stronger Foundations to Guide Research, Clinical Practice and Public Health. Pharmacy (Basel). 2019;7(3). 2019;
Reference number 5: there is no review yet, but more recent studies, as Chang et al. Polypharmacy, hospitalization, and mortality risk: a nationwide cohort study. Sci. Rep. 10(1) 2020 or Schöttker et al. Updated analysis on polypharmacy and mortality from the ESTHER study. Eur. J. Clin. Pharmacol. 74(7): 981-982 2018.
Reference number 14 refers to a Taiwan study, not comparable to the population of Spain
Response 12: Thanks for the suggestions. We have changed References number 1, 5 and 14.
Reviewer 2 Report
This paper describes epidemiological patterns of multimorbidity and polypharmacy in a large Spanish population identified by factor analysis on 2 occasions 4 years apart. The authors note that the patterns are mostly consistent between 2011 and 2015 with some changes that they mostly attribute to changes in medical record keeping. They conclude that consistency of associations between drugs and diseases over time could help in the early identification of potential interactions in multimorbid patients with high risk of negative health outcomes due to polypharmacy
The merits of this paper include the large dataset, carefully conducted analysis and description of multimorbidity and polypharmacy in children and adults <65 years. The epidemiological patterns identified are recognisable and make sense
Questions and comments for the authors:
- The authors define multimorbidity as the coexistence of 2 or more chronic diseases. Did the authors attempt to limit the diagnoses in the analysis to chronic diseases or the drugs to those with repeat prescriptions? The inclusion of acute respiratory infections in one of the factors indicates that they didn’t. How did these acute diagnoses/drugs influence their clusters? This is important if these patterns are meant to assist in preventing polypharmacy as people have many acute illnesses/short treatment courses throughout their lives which do not contribute to polypharmacy
I found the relatively high prevalence of multimorbidity (20-30%) and number of drugs taken (~2.5) in 0-14 year olds surprising. Does the inclusion of acute conditions in the analysis explain this high prevalence? How does this compare to the analysis of long term conditions in other paediatric populations?
- The authors claim (line 353) that this is ‘evolution of multimorbidity and polypharmacy patterns between 4 years in the same population’. It wasn’t clear to me whether 4 years was sufficient to see evolution and how the unidentifiable flux in the population accounted for any changes in patterns over time. I was actually not clear what the analysis of the 2011 data added to that already obtained from analysing the 2015 data.
- There are multiple other studies looking at multimorbidity clusters. The authors pay little attention to these beyond a very brief paragraph in the discussion that says their study is more comprehensive. A brief comparison of their patterns to others identified would be of merit
- The number of drugs taken by all age groups decreases between 2011-2015 – do the authors have any idea why this might be the case?
- I found the discussion quite long – it could be more focussed. It also went into quite a lot of detail about drug interactions, which was not merited by the analysis or described in the results and felt beyond the remit of the paper. The potential implications of this descriptive analysis were overcalled by the conclusions in the discussion and abstract
Author Response
Response to Reviewer 2 Comments
Point 1: This paper describes epidemiological patterns of multimorbidity and polypharmacy in a large Spanish population identified by factor analysis on 2 occasions 4 years apart. The authors note that the patterns are mostly consistent between 2011 and 2015 with some changes that they mostly attribute to changes in medical record keeping. They conclude that consistency of associations between drugs and diseases over time could help in the early identification of potential interactions in multimorbid patients with high risk of negative health outcomes due to polypharmacy.
Response 1: Dear Reviewer, thanks for evaluating our work and for your suggestions.
Point 2: The merits of this paper include the large dataset, carefully conducted analysis and description of multimorbidity and polypharmacy in children and adults <65 years. The epidemiological patterns identified are recognisable and make sense
Response 2: Thanks, we really appreciate your considerations on our work.
Point 3: The authors define multimorbidity as the coexistence of 2 or more chronic diseases. Did the authors attempt to limit the diagnoses in the analysis to chronic diseases or the drugs to those with repeat prescriptions? The inclusion of acute respiratory infections in one of the factors indicates that they didn’t. How did these acute diagnoses/drugs influence their clusters? This is important if these patterns are meant to assist in preventing polypharmacy as people have many acute illnesses/short treatment courses throughout their lives which do not contribute to polypharmacy.
Response 3: We considered demographic variables (i.e., age and sex), diagnoses of chronic diseases from primary care and hospitals, and dispensed drugs during 2015 from pharmacy billing records.
Diagnoses were originally coded according to the International Classification of Primary Care (ICPC) and to the International Classification of Diseases, 9th Revision (ICD-9), and were grouped in the Expanded Diagnostic Clusters (EDC) of the Johns Hopkins ACG System (version 11.0, The Johns Hopkins University, Baltimore, MD, US). All 114 diseases classified as chronic by Salisbury et al (1) were included in the analysis and coded in binary format (i.e., absence/presence of the disease). Additionally, we included rhinitis, according to the recent World Health Organization (WHO) indications (2), and acute lower respiratory tract infection, as it can lead to chronic sequelae. No other acute diseases were included.
Drugs were coded according to the Anatomical Therapeutic Chemical Classification (ATC) System at the third level to facilitate data processing, also in binary format. We included chronic and acute dispensation that had a prevalence of at least 3% in 2015.
We have explained it in more detail to make it clearer. Please, see lines 86-87 and 89-90, page 2.
Point 4: I found the relatively high prevalence of multimorbidity (20-30%) and number of drugs taken (~2.5) in 0-14 year olds surprising. Does the inclusion of acute conditions in the analysis explain this high prevalence? How does this compare to the analysis of long term conditions in other paediatric populations?
Response 4: The inclusion of rhinitis (3), according to the recent World Health Organization (WHO) indications and acute lower respiratory tract infection, as it can lead to chronic sequelae could explain this prevalence because these processes are very frequent in this period of age. Also, the exhaustive list of 114 chronic conditions used to define multimorbidity could partially explain the high prevalence, as other studies only take into account a smaller number of diseases.
The high prevalence has been found in other studies. In a representative sample of Swiss primary care patients, a substantial part shows multimorbidity with a high prevalence of chronic diseases, multiple drug treatment, and care dependency (4).
Point 5: The authors claim (line 353) that this is ‘evolution of multimorbidity and polypharmacy patterns between 4 years in the same population’. It wasn’t clear to me whether 4 years was sufficient to see evolution and how the unidentifiable flux in the population accounted for any changes in patterns over time. I was actually not clear what the analysis of the 2011 data added to that already obtained from analysing the 2015 data.
Response 5: We understand your concern and we certainly considered that a time span of four years cannot detect long-term changes in the same population. Reason why we expect to repeat this study by analysing even longer periods. Nevertheless, this study aims to compare the baseline epidemiological patterns of multimorbidity and polypharmacy in a given population by analysing the nature and changes of clinical clusters also analysing differences in drugs prescribed and diseases diagnosed which composed each cluster. So, this study does not add some information to that of 2015, rather is focused on investigating the presence of systematic associations between diseases and drugs at two different times, founding that most clinical profiles were maintained over time, such as mental, cardiometabolic, mechanical, endocrinological and osteometabolic patterns and their components.
However, the mental pattern did not occupy the first positions in 2011, unlike in 2015. These studies allow us to see the evolution in the patterns of use of drugs associated with diagnoses over the years, which is influenced by the commercialization of new drugs. For example, the use of pregabalin increased 75% between 2007-2013 (5).
We have explained it in more detail to make it clearer. Please, see Lines 449-450, page 22.
Point 6: There are multiple other studies looking at multimorbidity clusters. The authors pay little attention to these beyond a very brief paragraph in the discussion that says their study is more comprehensive. A brief comparison of their patterns to others identified would be of merit
Response 6: Thank you for your observation. Our main objetive was to to compare the baseline epidemiological patterns of multimorbidity and polypharmacy of the EpiChron Cohort in 2011 with those published in 2015 and to describe the clinical evolution of the clinical clusters identified. In addition, we tried to compare this study with other publications, but in the bibliography we observed many studies on multimorbidity patterns and others on polypharmacy patterns. There are few studies that connects multimorbidity and polypharmacy patterns at the same time. We have rephrased this paragraph to better compare our study with others: Line 364-376.
Point 7: The number of drugs taken by all age groups decreases between 2011-2015 – do the authors have any idea why this might be the case?
Response 7: Thank you very much for this interesting appreciation. When we performed the data analysis, we also observed this unexpected difference.
The decreasing number of drugs between the two years could be explained by two reasons. Firstly, in January 2012, the Health System of Spain stopped financing multiple pharmaceutical products such as expectorants, cough suppressants, propulsives and capillary stabilizing agents, so we did not have the pharmacy data for this type of medication. But it did not mean that they were not being used. We explained this difference in line 265-267.
Secondly, in April 2012 an increase in the copayment was included in the pharmaceutical prescription based on the income level, increasing the percentage of copayment significantly, which had a great impact on the decrease in the use of medications (6).
For these reasons, there was more prescription of drugs financed by the Health System in 2011 than in 2015.
Point 8: I found the discussion quite long – it could be more focussed. It also went into quite a lot of detail about drug interactions, which was not merited by the analysis or described in the results and felt beyond the remit of the paper. The potential implications of this descriptive analysis were overcalled by the conclusions in the discussion and abstract
Response 8: Our discussion could be considered long but it explained the patterns in 2011 and 2015 and made a comparision between them by age and sex, so it is difficult to summarize all this important information. We made an explanation of each of the associations because the main aim of this kind of cluster studies is to describe how the different chronic diseases and drugs are related and change according to age and sex. Understanding the mechanism of the relationship between them can lead to establishing certain preventive measures.
We have focused a little more on the pharmacological issue, due to the importance that drug interactions and therapeutic cascade have on the quality of life of patients. This fact has been observed in the study on the safety of patients in Primary Health Care (APEAS study). In this study, it has been seen that 48.2% of adverse events (adverse effects, lack of adherence, monitoring insufficient and wrong medications) in Primary Care were due to medication and 59.1% of these were avoidable (7). That is why we considered that the description of these interactions, adverse reaction and prescribing cascade that we observed is very important because it influences in the quality of life of patients.
- Salisbury C, Johnson L, Purdy S, Valderas JM, Montgomery AA. Epidemiology and impact of multimor- bidity in primary care: a retrospective cohort study. Br J Gen Pract. 2011; 61(582):e12–21. https://doi. org/10.3399/bjgp11X548929 PMID: 21401985
- Organization; WWH. WHO | Chronic respiratory diseases (CRDs). WHO [Internet]. 2015; Available from: http://www.who.int/respiratory/en/
- Allergic Rhinitis and its Impact on Asthma (ARIA) Phase 4 (2018): Change management in allergic rhinitis and asthma multimorbidity using mobile technology. J Allergy Clin Immunol. 2019 Mar;143(3):864-879.doi: 10.1016/ j.jaci.2018.08.049.Epub 2018 Sep 29.
- Gnädinger M, Herzig L, Ceschi A, Conen D, Staehelin A, Zoller M, Puhan MA. Chronic conditions and multimorbidity in a primary care population: a study in the Swiss Sentinel Surveillance Network (Sentinella). Int J Public Health. 2018 Dec;63(9):1017-1026. doi: 10.1007/s00038-018-1114-6. Epub 2018 May 21. PMID: 29786762; PMCID: PMC6245242.
- AEMPS, Use of opioid drugs in Spain during the 2008-2015 period. Available in:https://www.aemps.gob.es/medicamentosUsoHumano/observatorio/docs/opioides-2008-2015.pdf
- Real Decreto-ley 16/2012, de 20 de abril, de medidas urgentes para garantizar la sostenibilidad del Sistema Nacional de Salud y mejorar la calidad y seguridad de sus prestaciones. «BOE» núm. 98, de 24 de abril de 2012, páginas 31278 a 31312 https://www.boe.es/eli/es/rdl/2012/04/20/16
- Estudio sobre la seguridad de los pacientes en Atención Primaria de Salud (estudio APEAS)- Ministerio de Sanidad, Consumo y Bienestar. https://www.mscbs.gob